# Visually guided homing of bumblebees in ambiguous situations: A behavioural and modelling study

**Charlotte Doussot** *, **Olivier J. N. Bertrand**, **Martin Egelhaaf**

Neurobiology, Faculty of Biology, Universität Bielefeld, Germany

* charlotte.doussot@uni-bielefeld.de

**Data Availability Statement:** The data and codes are available in the following repository: DOI 10.4119/unibi/2946065 (https://pub.uni-bielefeld.de/record/2946065).

## Abstract

Returning home is a crucial task accomplished daily by many animals, including humans. Because of their tiny brains, insects, like bees or ants, are good study models for efficient navigation strategies. Bees and ants are known to rely mainly on learned visual information about the nest surroundings to pinpoint their barely visible nest-entrance. During the return, when the actual sight of the insect matches the learned information, the insect is easily guided home. Occasionally, modifications to the visual environment may take place while the insect is on a foraging trip. Here, we addressed the ecologically relevant question of how bumblebees' homing is affected by such a situation. In an artificial setting, we habituated bees to be guided to their nest by two constellations of visual cues. After habituation, these cues were displaced during foraging trips into a conflict situation. We recorded bumblebees' return flights in such circumstances and investigated where they search for their nest entrance following the degree of displacement between the two visually relevant cues. Bumblebees mostly searched at the fictive nest location as indicated by either cue constellation, but never at a compromise location between them. We compared these experimental results to the predictions of different types of homing models. We found that models guiding an agent by a single holistic view of the nest surroundings could not account for the bumblebees' search behaviour in cue-conflict situations. Instead, homing models relying on multiple views were sufficient. We could further show that homing models required fewer views and got more robust to height changes if optic flow-based spatial information was encoded and learned, rather than just brightness information.

## Author summary

Returning home sounds trivial, but to a concealed underground location like a burrow, is less easy. For the buff-tailed bumblebees, this task is a routine. After collecting pollen in gardens or flowered meadows, bees must return to their underground nest to feed the queen's larvae. The nest entrance is almost invisible for a returning bee; therefore, it guides its flight by information about the surrounding visual environment. Since the seminal work of Timbergern, many experiments have focused on how visual information is

**Funding:** The project is funded by the Deutsche Forschungsgemeinschaft (DFG). The funder had no role in study design, data collection and analysis, decision to publish, or preparation of the manuscript.

**Competing interests:** The authors have declared that no competing interests exist.

guiding foraging insects back home. In these experiments, returning foragers were confronted with a coherent displacement of the entire nest surroundings, hence, leading the bees to a unique new location. But in nature, the objects constituting the visual environment maybe unorderly displaced, as some are differently inclined to the action of different factors, e.g. wind. In our study, we moved objects in a tricky way to create two fictitious nest entrances. The bees searched at the fictitious nest entrances, but never in-between. The distance between the fictitious nests affected the bees' search. Finally, we could predict the search location by using bio-inspired homing models potentially interesting for implementing in autonomous robots.

## Introduction

Returning home, often termed homing, refers to the process of navigating from a location, such as a food source, back to the home surroundings. Even animals with tiny brains, such as bees, wasps or ants, accomplish this complex task daily. These hymenopteran insects may travel large distances to collect food and then return to their home surroundings by using a combination of different navigational strategies. For example, they can follow a visually familiar route [1–4], a scent trail [5, 6], or combine directional information from a compass (e.g. polarised light [7], sun or moon position [8, 9]) with the distance travelled as obtained from path integration [10, 11]. But, even when they arrive in the vicinity of the nest, the entrance may remain inconspicuous. In the case of the bumblebees, *Bombus terrestris*, the nest entrance is usually a small hole in the ground often hidden by vegetation. Therefore, to locate the nest hole, the insect performs a specific behaviour called local homing.

From the analysis of local homing by ground-nesting hymenopterans, including *Bombus terrestris*, it is known that when trying to find the nest hole, they rely primarily on sensory information such as visual, olfactory, or tactile cues [12–18]. However, among the senses that bumblebees and other hymenopterans have, vision plays an essential and dominant role in local homing since these insects are provided with an almost panoramic field of view [19–23].

On their first exit flights from their nest, bumblebees may learn a plethora of visual cues surrounding the nest hole, for instance, the overall skyline [24–27], the bearing of objects [28, 29], or the brightness of the panorama [30]. These learned visual cues are then used by the bees to guide their return flights to the nest. For example, if an insect was accustomed to reaching a location indicated by very conspicuous objects in its vicinity, after moving these objects to a new location, the bees will look for their nest at this newly designated site [12, 31–33].

However, in natural situations, environmental changes may be much less systematic; for example, small objects may be moved by the wind to a neighbouring location, while larger objects may stray at their place or may be moved only little. The displacements of environmental objects may then result in a new geometric configuration. If such changes occur while the insect is foraging at more distant sites, the visual environment around the nest hole experienced on its return will be different from the one stored in the memory. In this situation and in the absence of other directional cues, the displaced objects in the nest environment will now indicate to the bee different possible nest locations. Therefore, there will be a conflict as to where the nest should be located. How will the bumblebees behave in this kind of ambiguous situation? Will they search at the location indicated by one cue or the other, or will they look at a place of compromise between the different cues?

To answer these questions, we conducted behavioural experiments with bumblebees. In a flight arena, the bees were accustomed to returning home with the entrance to their nest being

surrounded by two constellations of objects: three large stripes on the arena wall and three small cylinders closely surrounding the entrance. During the tests, we moved the two landmark constellations in relation to each other and to the nest. Then, we recorded where in the flight arena the bees were looking for their nest.

Visually guided local homing of insects, without cue conflicts, has been successfully reproduced by several models [30, 34–38]. These models allow an agent (i.e. a simulated insect) to return home using views of the home surroundings collected during the first trips outside the nest. There is a broad consensus that a single or multiple snapshots of the landscape acquired at or near the nest entrance can guide a modelled bee to its home, at least if the home surroundings are kept stable between learning and return [30, 36]. However, do these models predict where the bees will look for their nest in ambiguous scenarios? To date, the performance of visual guidance models has not yet been systematically studied and compared to the behaviour of insects in such a scenario.

To understand how visual guidance of bumblebees in ambiguous situations is achieved, we proceeded in four steps. First, we studied where bumblebees search for their nest in visually ambiguous situations. Second, we selected models that allowed homing in the absence of conflict, i.e. before the cues were moved. Third, we simulated where bees would search for their nest in visually ambiguous situations based on these models. Finally, these predictions were compared with the experimentally determined search locations of bumblebees. By this combined behavioural and modelling analysis, we found that only models using multiple views account for the behaviour of bumblebees under visually ambiguous conditions.

## Materials and methods

### Experimental design

We used three healthy hives of *Bombus terrestris* provided by Koppert B.V., The Netherlands. The bees had direct access to pollen in the hive. The hive was connected with a transparent tubing system to a large cylindrical flight arena (of 75 cm radius and 90 cm height) (Fig 1A). The bumblebees entered the arena through a 1 cm hole in its floor. The entire arena floor was covered with small wood chips to hide the nest hole efficiently; they were frequently shuffled around to avoid bumblebees to use potential odour cues set by others. The arena was covered with two acrylic plates (not shown in Fig 1A) to prohibit the bees from escaping the flight arena. The arena wall was divided into a bottom part of 80 cm height, which could be rotated around the vertical axis of the arena, and an upper fixed part of 10 cm height (Fig 1A). The bumblebees could leave the arena and access a foraging chamber through a 1.5 cm diameter hole in the fixed part, where feeders provided them with a sweet aqueous solution (a mixture of 30% saccharose and honey drops).

In the flight arena, two conspicuous constellations of visual cues were provided to the bumblebees. The first constellation consisted of three black cylinders (15 cm height and 2 cm diameter each), arranged at 10 cm distance around the nest hole. The second was a pattern of three red stripes on the white wall of the flight arena (80 cm height and 12 cm width). These background stripes were asymmetrically arranged, with two of them placed next to the nest hole (Fig 1B). The bars were red for tracking purposes (bumblebees are not sensitive in this spectral range and would perceive them as dark [39]). A white mesh cloth covered the ceiling of the room to restrain access to external cues (Fig 1A).

A bumblebee flying from the foraging chamber back to the hive was recorded by two synchronized high-speed cameras (Falcon2 4M, Teledyne DALSA, Inc.) with a resolution of 2048*2048 pixels at 74 frames per seconds. The two cameras were viewing the set-up at different angles, giving a top view and a tilted view of the arena. Ten LEDs (OSRAM, 350 lm),

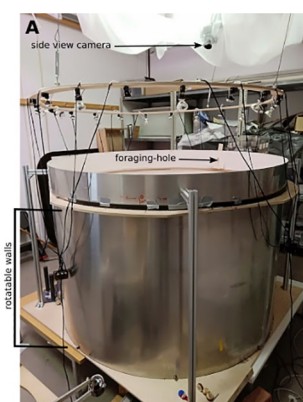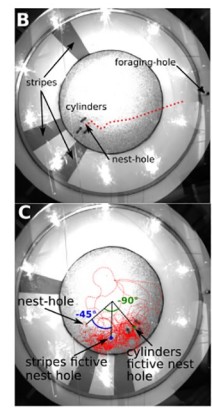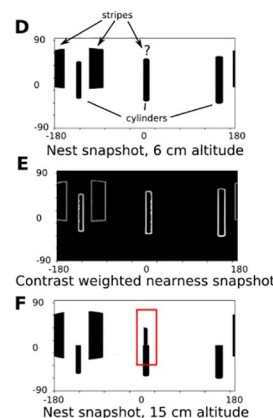

**Fig 1. Experimental set-up.** A: Photograph of the experimental set-up consisting of a cylindrical flight arena, a suspended ring holding part of the lighting, and a white mesh covering the camera holders (only the side camera is visible in the picture). B: Habituation condition, image from the top view camera during recording, with two of the three stripes on the arena wall placed behind the nest hole and three cylinders placed around the nest. Overlaid trajectory of a return flight during the habituation phase, red dotted line. C: Picture of a test condition, cue conflict condition -45/-90; the cylinders' fictive nest, green circle, is placed at an angle of -90° from the nest hole viewed from the centre of the arena, then, the stripes' fictive nest, blue circle, at -45° from the nest hole. The picture is overlaid by one sample trajectory corresponding to this cue conflict condition (red dotted line). D: Nest snapshot, equirectangular projection of the simplified bee view, taken at 6 cm above the nest hole location in the simplified rendered environment. Note that the third bar is occluded by one cylinder. E: Image of the contrast-weighted nearness nest-snapshot in the rendered environment at 6cm altitude. F: Nest-snapshot at an altitude of 15cm. The third bar is now only partly occluded.

mounted on a wooden ring above the cylinder, illuminated the arena, and 16 paired neon tubes (Biolux 965, Osram, Germany) arranged symmetrically in the room served as additional light sources.

We started the recordings as soon as a bumblebee entered the flight arena. Recordings lasted up to 5 minutes or until the bee found the nest entrance. From the two calibrated cameras (Matlab toolbox DLTdv 5, Hedrick's lab), we tracked the position of the bumblebees with a custom-made Python script, based on OpenCV. The videos were then manually reviewed with the software IVtrace (https://opensource.cit-ec.de/projects/ivtools), so the bee's position could be manually corrected in case of a tracking error.

## Habituation and test procedure

We let the bees exploring the set-up for three days in the arrangement shown in Fig 1B. A naturalistic day-night-cycle was reproduced by having lights on between 7 am and 7 pm, and off otherwise. The bees flew only during the light time. After this exploration and learning phase, relatively straight trips from the foraging chamber back to the nest entrance were observed (see Results & Fig 1B). To test how the bees behaved under a visually ambiguous condition, we trapped at least five foragers while these were feeding in the foraging chamber. Then, while all other bees were constrained in the hive, the two types of visual cues in the flight arena were put into conflict by rotating the wall and displacing the cylinder constellation i.e. creating a cue-conflict (Fig 1C). After this manipulation, we allowed a single bee to re-enter the flight arena from the foraging chamber and recorded its flight. In the rare case of bumblebees not showing any attempt to find their nest (i.e., never flew close to the ground as they usually do when searching for the nest), the videos were not analysed. A searching bee had up to 5 minutes to find its nest. If it did not find the nest in this time interval, it was caught and placed manually back into the hive. The experiments were performed in the late morning for up to

two hours until all trapped bees were individually released in the flight arena and tested. Then, the cylinders and stripes were placed in the arena at their original position, so bees could fly in the non-conflict situation (Fig 1B) until the next set of experiments on the following day, giving them enough time to habituate back to this original situation. Because our bees were not marked and the number of foragers in a small bumblebee hive is limited (between 5 and 10), individual's replication could not be avoided. However, by blocking a minimum of 5 bees in the foraging chamber, which were then individually released in the test arena, we could assure at least five different individuals tested per condition and a time interval of approximately 22 hours between a possible individual replication. Hence, in the various figures, n = indicates the number of flights recorded and not the number of different individuals.

Several conditions were tested, all shown in Results. Test situations could create two possible locations for the nest hole, i.e. two fictive nest holes, relative to either one or the other type of cues (stripes or cylinders). The different conditions are described by three numbers (e.g. -90/-45, -45˚, Fig 1C). The first number indicates the angle, as seen from the arena centre, between the real and fictive nest hole determined by the cylinders. The second one shows the angle between the real and fictive nest hole defined by the stripes. The third number indicates the directed conflict angle between the two fictive nest holes. In total, 12 cue-conflict conditions were tested.

## Behavioural analysis

During the 5 minutes tests, bumblebees usually flew at low altitudes in the arena searching for their nest hole in the ground. We first estimated at which altitude bumblebees spent most time by analysing their distribution along the arena z-axis (i.e. the altitude axis) (Fig 2). We observed that bumblebees spent 75 percent of their time at flight heights below 20.63 cm. Hence, an upper threshold of 20.63 cm was used to exclude behavioural sequences not related to the search for home. Besides, we used a lower limit of 3 cm to exclude walking behaviour of the bumblebees.

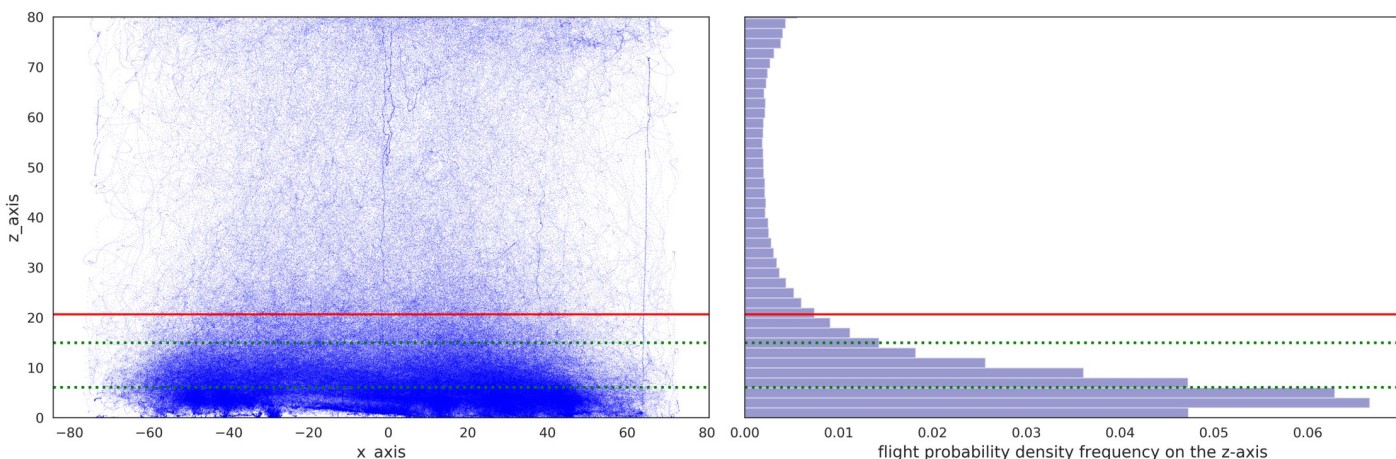

**Fig 2. Overall spatial distribution of all flights of bumblebees in the arena under cue-conflict conditions.** A: representation of all the flights obtained under cue conflict conditions (n = 107 flights) along the x-axis and z-axis inside the arena. The bee's position at each point in time is represented as a blue dot. Below the red line 75% of the flight time takes place. The two green dotted lines are the two altitudes for which a set of images has been taken from the perspective of a bee (6cm and 15cm). While searching for the nest hole in cue conflict situations bumblebees fly relatively close to the ground. Nevertheless, numerous data points are visible at higher altitudes; this is due to bumblebees abandoning their search or colliding with the perspex lid at 90 cm above the ground. B: Histogram representing the frequency distribution of flights along the z-axis of the arena.

To investigate where the bumblebees flew searching for their nest either at the two constellations or at a compromise location, we estimated the probability density function of the bee's location along the x-y-plane of the arena (with z between 3cm and 20.63 cm) by a 2D kernel density estimation (KDE, Python library Scipy.stats). The KDE bases its estimation on a smoothing parameter (estimated here thanks to the Scott's rule [40]) (see Results). Because some bumblebees find the nest in less than five minutes, we normalized the distribution for each flight, expressing the distribution as a proportion of the time spent in the arena. All time-normalised flight distributions from the same condition are summed. Then, this sum is divided by the number of flights (n) recorded in this condition, hence, leading to an average distribution. By doing so, the average distribution obtained is not affected by the different n.

## Homing models

We seek a computationally parsimonious model predicting the search locations of bees in our cue-conflict experiments. Hence, we simulated bees following different homing models for the environmental conditions of our experimental analysis. The simulated bees, later called agents, first gather information about the nest surroundings based on panoramic views.

We rendered a simplistic version of the set-up in a graphic software (Blender, Version 2.79), and considered only potentially relevant visual cues, i.e., the arena wall with the stripes and the cylinders. So, the entrance to the arena from the feeding chamber and the nest hole were not modelled to focus the model comparisons only on the conflicting cues (Fig 1D). Consequently, some cue-conflict conditions were visually identical in the rendered arena (e.g., cylinders/stripes: 45°/-135° and 90°/-90°). Hence, the set of conditions to be tested was reduced to only seven see Results. From the rendered arena, we took a series of panoramic images oriented along the x-axis (i.e. with the azimuthal null viewing direction along the x-axis of the arena), spaced on a grid by 2 cm, making a total of 7211 views. Since bees flew for most of the time below 20.63 cm (see above), the rendering procedure was done on two grids at different altitudes: 6cm (approximating the median value of the bees' distribution along the height of the arena, see above), and 15 cm (the mid-altitude of the fourth quantile)(Fig 3B). The gathered images are equirectangular panoramic snapshots (-180 to 180° in azimuth and -90 to 90° in elevation, with a resolution of 1px per degree). Based on these sets of rendered images, different vision-based homing algorithms were tested, that relied on one or multiple panoramic snapshots of the nest environment. Additionally, for the sake of comparison a homing algorithm based on explicit knowledge of the cue positions was tested (Fig 3B, Table 1). All homing algorithms used the memorized representation of the habituation condition (Fig 1B & 1D), i.e. the situation without cue conflict.

Homing models rely on a memory encoding visual information linked to the home location to allow the agent to find home. When returning home, they also need a method to compare this memory with the visual information about the actual surroundings. This comparison leads to a home vector. The home direction is encoded in the home vector's argument. We calculated the homing vector at each grid location to determine the agent's movement direction anywhere in the arena. Consequently, each model yields a vector field.

Numerous variants of homing models have been described and tested in the last decades, be they using frequency-based [41], rotation invariant [26], brightness [30], skyline [27], or optic-flow representations [42], single snapshot [30], multi-snapshots [36, 43], or attractive and repulsive views [38]. We sought first for parsimonious models to predict the bumblebees' search location in visual conflict situations. We only wanted to increase the computational complexity of the models if none of the simpler models would be able to explain the

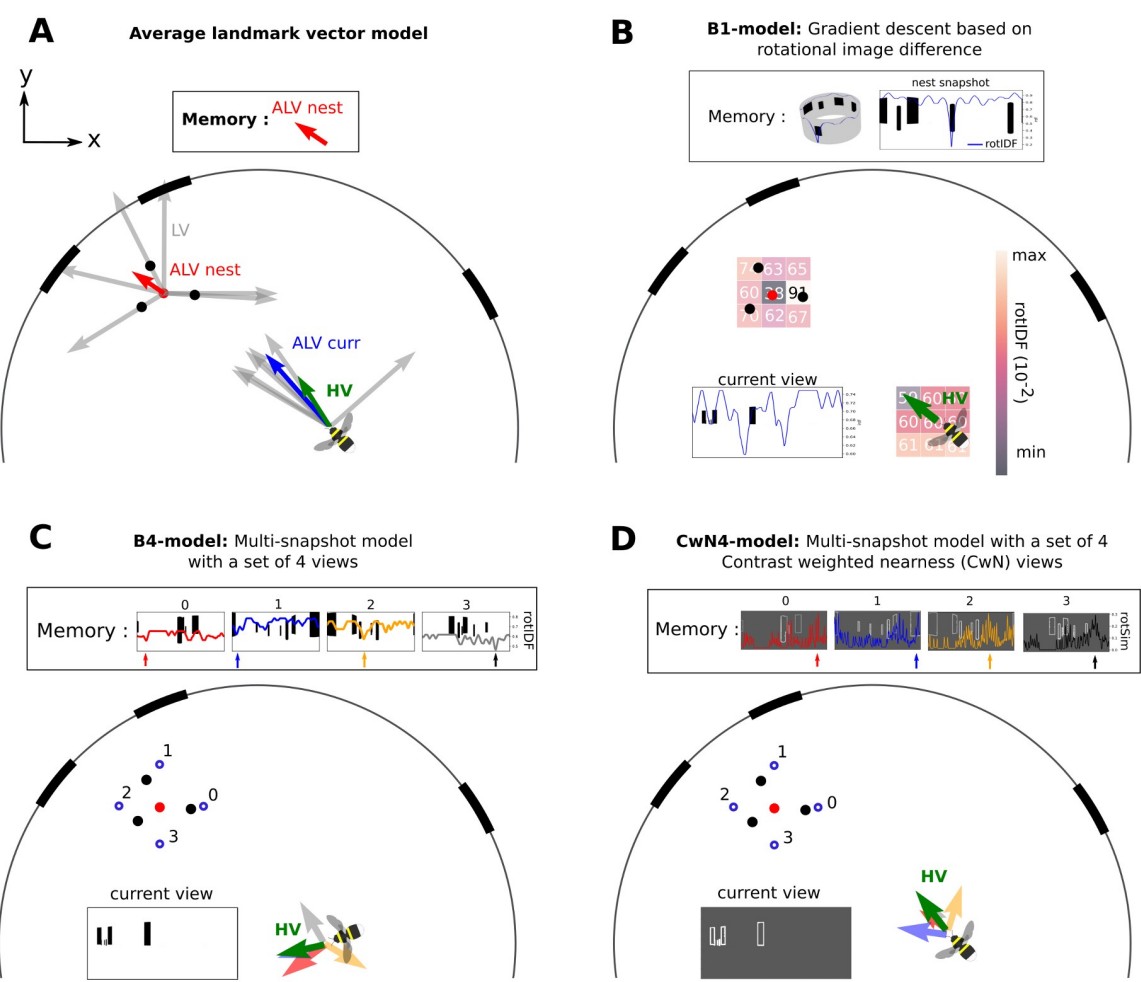

**Fig 3. Schematic of the different tested models.** A: Average landmark vector model, or ALV model: The stored vector in the memory is the ALV at the nest hole location (red arrow). The ALV at the nest is the average of each landmark vector (LV) at this location (grey arrows). The ALV at the current location is calculated in the same manner (blue arrow), and the ALV-nest is subtracted from this one. The resulting vector is the homing direction or Home Vector (HV) represented with a green arrow. B: the gradient descent model based on the image rotational difference function (rotIDF) of brigthness values will be termed B1-model; the minimum rotIDF is calculated at each grid position, the potential obtain is plotted on a downsampled grid and a colourmap (from black to white), indicating the value of the minimum rotIDF. The agent will descent the potential, thus, moving in the direction of the darkest neighbouring point on the represented grid. The rotIDF is overlaid with the current view by a blue line. The nest snapshot stored in memory shows a null minimum in the rotIDF at the middle of the image for 0° orientation. Important: the value of the minimum rotIDF at the current location is the only information used to create the potential. Finally, the minimum rotIDF values on the represented coloured grid is not null at the nest due to the downsampling of the grid for illustration purposes, the grid is not falling below the exact nest location where the minimum rotIDF is indeed null. C: The multi-snapshot model or Bn-model. In this example, 4 views constituting a set of views are taken outside of the cylinder constellation on a 15 cm radius circle (B4-model). Each view is overlaid by its rotIDF function with the current view. Each view refers to a heading direction of the same colour (red, blue, yellow and grey). The green arrow represents the weighted circular average of all headings, it is the HV. D: The multi-snapshot model based on contrast weighted nearness (CwN) views, or CwNn-model. 4 CwN snapshots at 15 cm from the nest are taken (CwN4-model). Each view is overlaid by its similarity function with the current view, leading to different heading directions at the current position (coloured arrows). The weighted circular mean of the different headings is the green arrow indicating the HV direction.

data. We will later discuss how our findings might pertain also to models not tested here (see Discussion).

In the following paragraphs, we describe the models tested in our study. The first model is a non-holistic (i.e. the memory encodes only information relative to visual landmarks that first need to be distinguished in the overall visual surroundings); it is a computationally cheap and

**Table 1. Main criteria distinguishing the different tested models: The need for a compass, the type of cue, its holistic nature, the memory and the mechanism used for comparison between the current visual information and the memory.** CwN stands for the contrast of the image weighted by the nearness, rotIDF for the rotational image difference function and the rotSimF for the rotational similarity function. Models original names are mentioned in the method section and Fig 3.

| Visually based homing models List | | | | | |
|---|---|---|---|---|---|
| Model acronyms | Requires compass | Holistic | Memory | Visual cues | Comparison method |
| ALV | yes | no | one vector | landmark bearing | vector substraction |
| B1-model | no | yes | panoramic snapshot | brightness values | rotIDF |
| CwN1-model | no | yes | panoramic CwN snapshot | distance and contrast | rotSimF |
| Bn-model | no | yes | several panoramic snapshots | brightness values | rotIDF |
| CwNn-model | no | yes | several panoramic snapshots | distance and contrast | rotSimF |

well-studied model and serves as an alternative to snapshot-navigation [29, 44]. The second and third models are based on holistic memories (i.e. the memory include the whole visual environment without segmentation into landmarks) of the visual scenery around the nest, i.e. a panoramic snapshot. The fourth and fifth models are based on holistic memories of the visual scenery around the nest i.e. multiple panoramic snapshots.

**The Average Landmark Vector (ALV Model).** To accomplish homing, the agent used as visual information the bearing of visual landmarks on its retina: in our case, each of the three cylinders and each of the three stripes. Hence, this model does not use the rendered views. This method implies the segregation of the environment between landmarks and non-landmarks, making this model non-holistic. Each bearing led to a unit vector, called a landmark vector and, thus, does not encode the distance to the landmark, $(\vec{LV})$ (Fig 3A). From these 6 $\vec{LV}s$ the agent calculated their average: the average landmark vector $(\vec{ALV})$. The $\vec{ALV}$ at the nest location was the vector kept in memory by the agent. The home vector $(\vec{HV})$ was the difference between its current average landmark vector at its location $(\vec{ALV}_{curr})$ and the memorized one $(\vec{ALV}_{nest})$. This model requires an external compass to perform meaningful vector calculations based on an x-y coordinate system. Thus, for the simulation, a perfect compass aligned with the grid indicating for the x and y-axis was used.

**Brightness-snapshot model (B1-model).** This model (Fig 3B) relies on the brightness value of each pixel in panoramic views of the environment as visual information [19, 30]. These views were not further segmented; thus, the agent used the environment as a whole (i.e. holistic). The agent memorized a single panoramic view taken at the goal location. Then, the agent used the difference between its current and the memorised view to compute the home vector at all grid locations. However, determining the image difference in a useful way is not an easy task; for example, two panoramic images acquired in different orientations but at the same location yield large differences. To solve this issue, one can rotate one image against the other until the difference is zero and the two images are aligned. Therefore, in a compass-free environment like ours, an agent not knowing its orientation in the arena could rotate to minimize the difference between its current view and the memorized view. Determining the minimum of the rotational image difference function (rotIDF) could be a reason why ants stop occasionally on their return trips to the nest and scan the environment on the spot before deciding where to go [45].

The minimum rotIDF, described by Eq (1) is the minimum of the square root of the average difference between the brightness values of the rotated current view ($I_{x,y}$ of axis $u$ azimuth and $v$, elevation; $I$ as dimensions of $(N_u, N_v) = (180, 90)$) for different azimuthal orientations, $\alpha$, and the nest snapshot ($I_N$). This calculation is also called the root mean squared difference (r. m.s) [30] (1). This formula had to be adjusted due to the distortion of the views caused by the

reprojection of the environment from a uniformly sampled 3D sphere, mimicking a bee's view, back to 2D equirectangular images. This distortion creates an oversampling at the poles. Consequently, a different weight, $w$, is applied to the image depending on the pixels' retinal position along elevation. This weight is given by a sine function along the image y-axis, resulting in a weight of 1 at the equator, and 0 at the pole as described by Eq (2) [42]. In this way, the pixels' values close to the poles have less weight for the computation. We add this weighting function (2) to the rotIDF formula (1).

From any point of the grid the agent had to descent along the gradient of the potential encoding the minimum rotIDF between the current view and the memorized view forming a vector field of home vectors (Fig 3B). Because this model uses only a single snapshot of brightness values to guide homing it is referred later in the text as the B1-model.

$$d_{x,y} = \min_{\alpha} \sqrt{\frac{\sum_{u,v} w(v)(I_{x,y}(u+\alpha, v) - I_N(u,v))^2}{N_u \sum_v w(v)}} \tag{1}$$

$$w(v) = \sin\left(\pi(v+0.5)/N_v\right) \tag{2}$$

**Multi brightness-snapshot model (Bn-model).**    The agent used the brightness values of panoramic views as visual information. It memorized several panoramic snapshots $s_i$, constituting a set of views $S = \{s_0; s_1; \ldots s_n\}$, located around the nest location. In the model, each snapshot was oriented toward the nest entrance. In nature, ground-nesting insects perform a learning choreography around their nest hole, during which they may collect views oriented towards the nest thanks to path integration [46–48], or the ability to visually track the nest hole at short distances [49]. When returning to its nest, the agent followed a homing vector calculated from the four rotIDFs between its actual view $s_{curr}$ and each memorized view in $S$. Therefore, a different heading direction at each grid location $x$, $y$ was determined for each snapshot Eq (3) based on the rotIDF. These headings were then weighted by the ratio between the minimum rotIDF (as computed in (1)) of all snapshots $s_i$ in $S_{dmin}$ (4) and the rotIDF of this one, $d_{s_i}$, as described in Eq (5). Finally, the homing vector $\vec{HV}$ was the weighted circular mean of the different heading directions $h_{s_i}$ (6). The homing vector depends on the set of memorized views; for example, the number of views in $S$ or their locations may affect the homing performance [36]. Therefore, we used different sets of views $S$. The different sets consisted of either 4 or 8 equally spaced views taken at two distances from the nest hole; at 5 cm (inside the cylinder constellation) or 15 cm (outside the cylinder constellation) (Fig 3C). Learning flights of bumblebees start in general close to the nest hole and gradually cover larger distances [50, 51]. By taking sets of views in the close vicinity of the nest hole, it is plausible that these views are collected by the bumblebees during learning. In addition, views were collected opposite to each other to increase the visual disparity, since this parameter could affect the homing success [36].

Because this type of model uses $n$ snapshots of brightness values to guide homing it is referred later in the text as the Bn-models.

$$h_{s_i,x,y} = \operatorname{argmin}(\operatorname{rotIDF}(I_{x,y}, s_i)) \tag{3}$$

$$S_{dmin} = \min_S(d_{s_i,x,y}) \tag{4}$$

$$w_{s_i} = \frac{S_{dmin}}{d_{s_i,x,y}} \tag{5}$$

$$H\vec{V} = \arg\left(\sum_S w_{s_i}.exp(h_{s_i,x,y}).i\right) \tag{6}$$

**Contrast-weighted-nearness-snapshot model (CwN1-model).** The agent used views encoding for the depth and contrast of the environment to perform its homing: contrast-weighted nearness views (CwN) (Figs 1E & 3D). The CwN map is calculated based on the Michelson contrast, i.e. the ratio of the luminance-amplitude ($I_{max} - I_{min}$) and luminance-background ($I_{max} + I_{min}$) within a 3x3 pixel window on the view. Then, the contrast was weighted by the inverse of the distance (nearness), obtained from the environment's spatial layout. Like the brightness-based model, the agent memorized the panoramic CwN view at the nest location. When returning home, the agent ascended the gradient of homing vectors following the maximum similarity between the CwN view at each grid point and the memorised view. The CwN map contains distance information. In nature, an insect gains distance information during translational movements through the array of elementary movement detectors (EMDs, for review [52]). The CwN acts as an approximation of the response profile of the insect's retiniotopic arrays of EMDs, as suggested by model simulations of these movement detector arrays [53, 54]. We applied the formula described by Dittmar et al. 2010 on each CwN grid-views ($x, y$) to calculate the rotational similarity function (rotSimF) between the current CwN view and the memorised view (7). The similarity is the correlation coefficient between the nest view ($I_N$) and the current view ($I_{x,y}$). As for the rotIDF, we assumed the agent to be able to internally rotate its current view to compute the best similarity value. Because this model uses only a single snapshot of CwN values to guide homing it is referred later in the text as the CwN1-model.

$$\text{sim}_{x,y} = \max_\alpha \frac{\sum_{u,v} w(v)(I_{x,y}(u+s,v)).I_N(u,v)}{\sqrt{\sum_{u,v} w(v)I_{x,y}(u,v)^2}\sqrt{\sum_{u,v} w(v)I_N(u,v)^2}} \tag{7}$$

**Multi contrast-weighted-nearness-snapshot model (CwNn-model).** The multi-snapshot model based on brightness views, as described above, was adapted here for CwN views and the rotSimF. Hence, the memory was a set of CwN views $S = \{s_0; s_1; ..; s_n\}$, where $s_i = CwN(x_i, y_i)$. Here, the agent followed an homing vector at each grid point computed by the weighted circular means of each headings of the CwN views in $S$ (Fig 3D). Because this type of model uses $n$ snapshots of CwN values to guide homing it is referred later in the text as the CwNn-models.

## Description of the homing algorithm behaviour using the Helmholtz-Hodge decomposition

From the vector fields (gradient) obtained by the different homing models, we wanted to infer where the agent was most likely to end its journey. This can be determined by studying where the vectors of the above-mentioned fields of homing vectors converge. We obtained the convergence of the vector field by decomposing it into two components using the Helmholtz-Hodge Decomposition. The component of interest here is the curl-free component, i.e. the divergence of a potential $\phi$ [55, 56]. This potential can be seen as basins, valleys, and summits: thus, we can conceive the agent as a fluid flowing from summits down to basins (Fig 5A). We described the agent's homing behaviour for each model by the topology of its potential, following this analogy the landscape was split along the z-axis at different levels, i.e. isohypses. The different isohypses

are shown with overlaid contour lines on Fig 5A. The highest isohypse surrounds a summit and the lowest a basin. Basins correspond to areas where the agent is homing. The creation of basins gives additional information like the size of the region of convergence. Thus, by using the Helmholtz-Hodge Decomposition, we could directly compare search areas, where the bees spent most of the time searching for their nest, with the basins' locations and shapes.

We ran the Helmholtz-Hodge Decomposition on each vector field for all homing models and then scaled the obtained potential between 0 and 1.

## Quantifying the models' predictions

The models gave predictions of where the animal may search for its nest, i.e. the basins' locations. Thus, we need to compare the basins' locations to the behaviour of the bees, i.e. the search areas. We defined search areas as regions where the probability density distribution of the bumblebees' location was above a certain threshold. This threshold was set at a third of the probability density function's maximum for each condition, splitting the areas between high probabilities and low probabilities of observing a search. Similarly, the homing landscape of the models was split between values above and below 0.15, where the models are more likely to converge, thus, corresponding to a basin. In this way, the homing landscape of the models was segregated into areas where search is predicted and areas where it is not predicted (respectively, the behaviour was split between search behaviour and non-search behaviour). From these predictions and observations we determined a confusion matrix (Table 2). From the confusion matrix, we calculated an accuracy measurement, the F1-score, of each model prediction (Eqs 8 to 10). The F1-score gives a good metric to check the accuracy of the models without ignoring the costly impact of false positives and negatives. Indeed, we consider false positives (*fp*) costly since the agent searches at a wrong location, which could theoretically impair its homing; false negatives (*fn*) show that the prediction fails to describe the full behaviour. The isohypse used to define the prediction areas is an important parameter, indeed, the lower the isohypse is set, the smaller the predicted search area gets. Consequently, we systematically varied isohypse values between 0.1 and 0.29 and studied its impact on the F1-score. We compared the different models' F1-score over a varying threshold for each condition using a Kruskal-Wallis test (because our data are not normally distributed), followed by a pairwise comparison using Dunn's test for multiple comparisons of independent samples. Finally, all p-values were adjusted by a Bonferroni correction.

$$precision = \frac{\text{tp}}{\text{tp} + \text{fp}} \qquad (8)$$

$$recall = \frac{\text{tp}}{\text{tp} + \text{fn}} \qquad (9)$$

**Table 2. The confusion matrix.**

| Behaviour | | Model prediction | | |
|---|---|---|---|---|
| | | Search prediction | Non-search prediction | Total |
| | Search | true positives | false negatives | *tp + fn* |
| | no-search | false positives | true negatives | *fp + tn* |
| | Total | *tp + fp* | *fn + tn* | *N* |

$$F1_{score} = 2.\frac{precision.recall}{precision + recall} \tag{10}$$

## Results

### Behavioural analysis

Bees fly straight back home in the familiar non-conflict situation after they enter the test arena from the foraging chamber (see examples Fig 4A). But how do they search for their home when visual cues surrounding their nest have been brought into conflict? After the two cue constellations formed by the stripes and the cylinders, respectively, have been moved relative to each other (i.e. visual conflict), bumblebees entering the flight arena from the feeding chamber, fly towards the ground and start searching for the nest hole. Their dedication to find home is reflected in the flights' height distribution in the arena (Fig 2). Interestingly, most bumblebees do not continuously search for their nest hole during the entire 5 minutes intervals. They sometimes fly at higher altitudes or even against the transparent ceiling of the arena. Where do bumblebees search for their nest when the different cue constellations indicate two different nest entrances? For a given condition most trajectories show similar search locations, but some variability is visible (S1 Fig). This variability is expressed by the time spent searching, and the area of the search. Therefore, to account for these differences, and to get an idea on the overall behaviour under the different conflict conditions, we determined the probability distribution of the bumblebees' location in the arena (Fig 4). The bumblebees spend most time in restricted areas including the different fictive nest holes, corresponding to the two types of cues. They do not search at a compromise location between the two fictive nest holes. However, in a few cases we observed that they fly in the direction of the third stripe, which was located during habituation further away from the nest(S1 Fig, 45/-135). For all conditions, bumblebees search at the stripes' fictive nest location. When the conflict between the cues is small, they mostly search at the cylinders' fictive nest (e.g. conditions for 45˚ and -45˚ conflict, Fig 4B). Hence, the probability density distribution seems to be influenced by the placement of the cues relative to each other.

### Simulations of visually based homing under non-conflict condition

Various homing models have been proposed to explain the behaviour of hymenopterans returning to their nest hole (see Introduction and Material and methods). We investigate in the present study the homing success of some of these models in our environment and, especially, under cue-conflict conditions Fig 1B and 1C.

Interestingly, not all the models, when scrutinizing the potential derived from their field of homing vectors in the arena, succeed to account for homing even in the non-conflict situation (Fig 5). Indeed, the lowest isohypse, indicating where the agent is led, does not always surround the nest location (Fig 5B, 5C and 5D). The two models (B1-model and CwN1-model) memorizing a single snapshot at the nest location, show very flat profiles and several basins at locations different from the nest hole, i.e. local minima. The only models fitting the non-conflict situation are the ALV model and the models using more than one snapshot (Bn-models and CwNn-models). We tested 4 and 8 snapshots, i.e. the B4- and B8-model as well as the CwN4- and CwN8-model.

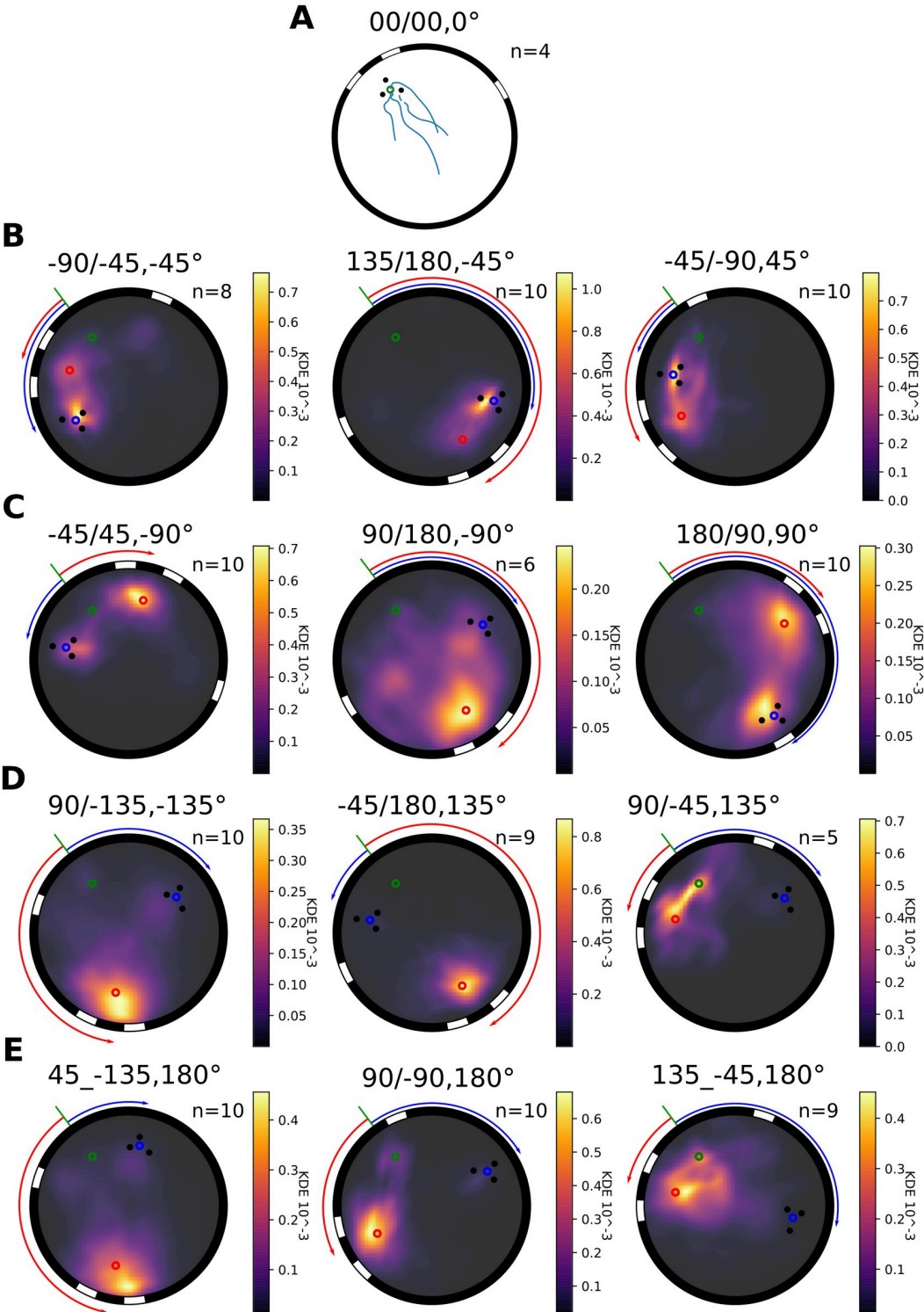

**Fig 4. Probability distribution of bumblebees' search location in the arena during conflict situations.** Each subplot represents the arena with the different cues; the three stripes and the three cylinders, and the corresponding fictive nest entrance; red and blue. The real nest hole is represented by a green dot. Rotations inflicted to the different cue-constellations are represented by an arrow of the corresponding colour. A: the trajectories of bumblebees during non-conflict situation (n = 4). For all the other subplots, the colourmap represents the normalized KDE of the flights distribution from low density (black) to higher (yellow). The number of flights in each condition is indicated in the corresponding sub-figures. Each sub-

title's numbers informs about the tested condition: the first number describes the angle between the real nest hole and the cylinders fictive nest hole from the centre of the arena, the second number informs of the stripes' fictive nest hole location. Finally, the last number describes the directed visual conflict angle between the two cues. B: 45˚ absolute visual-conflict conditions, C: 90˚, D: 135˚, E: 180˚.

The Bn-models (Fig 5C) lead to successful homing only when the snapshots are taken outside of the cylinder constellation. When these models use only four views, the lowest isohypse surrounds the centre of the arena rather than the nest hole, but this isohypse disappears when eight views are stored in the memory; thus, the agent will not be driven towards the nest hole but in the middle of the arena, if only four snapshots are used (Fig 5A and 5C). Also, when the views are collected inside the cylinder constellation, a basin is formed in the centre of the arena and not at the nest location. Besides, this basin persists even if eight snapshots taken

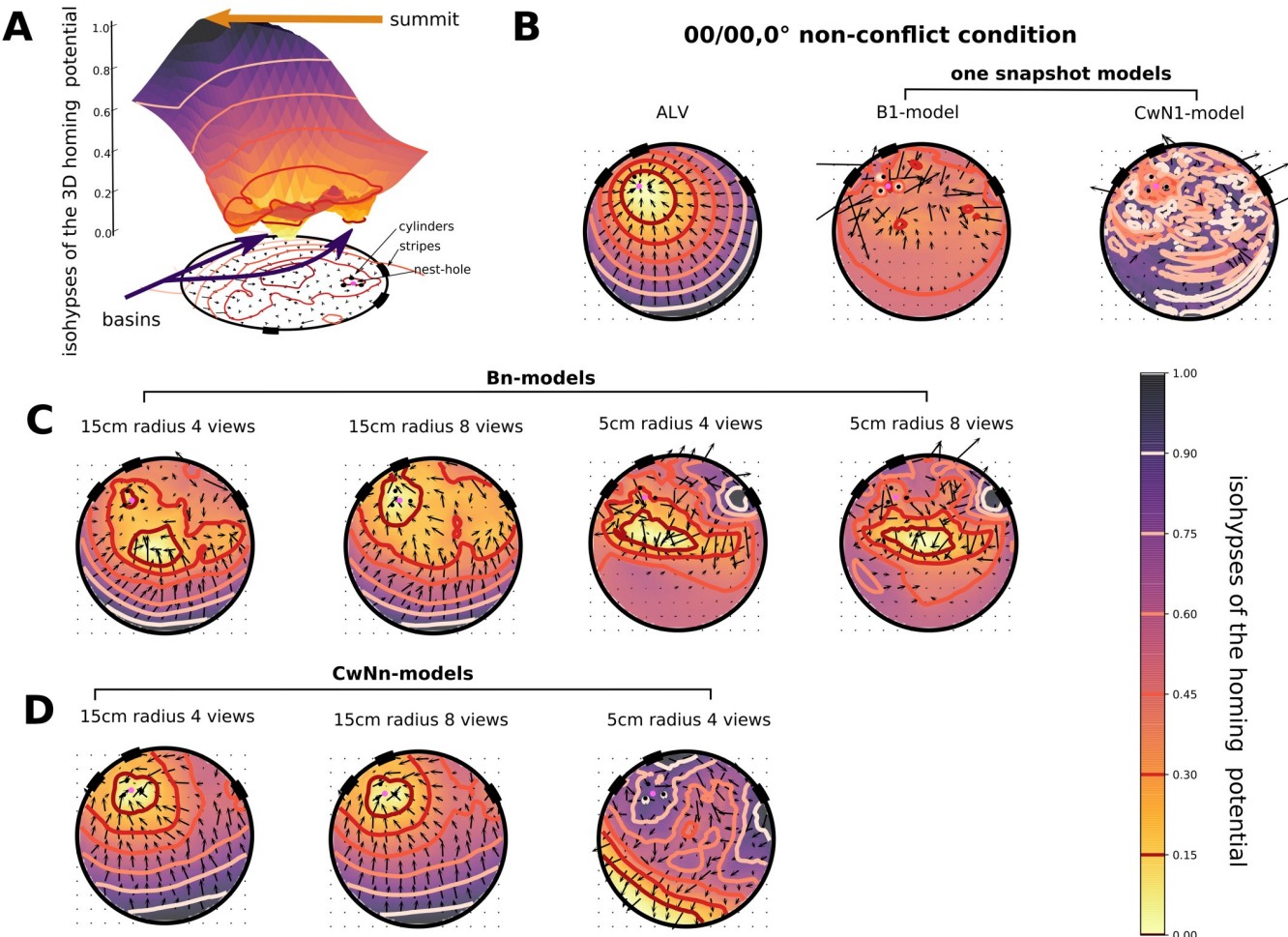

**Fig 5. Model homing potential during non-conflict situation (condition 00/00,0).** A: Helmholtz-Hodge Decomposition 3D profile of the B4-model, coloured along the z-axis, from yellow, basins, to black, summits, overlaid with contour lines indicating the different isohypses of the profile. Levels are projected on the ground with the schematic of the arena at the 00/00 condition. The nest hole is represented by a pink circle. In addition, the vector field from which the homing potential is derived is plotted. B: Performance of the ALV model homing and one-snapshot models (the type of model used is indicated above the individual plots), are described by the colourmap and the isohypses, overlaid with a vector field showing the homing vectors on each grid points of the downsampled grid. The performance of each model is plotted in the same manner. The yellow of the colourmap and the dark red of the contour indicates areas where the vector field is converging. C: Bn-models using 4 or 8 views taken at 15 cm or 5 cm from the nest (indicated above the plots). D: CwNn-models, based on 4 and 8 views taken at 15 cm from the nest, then 4 views taken at 5 cm from the nest.

inside the cylinder constellation are used. Hence, the agent is systematically driven towards the middle of the arena and unable to home successfully when views are taken within the cylinder constellation.

With the CwNn-models (Fig 5D) four views taken outside of the cylinder constellation were enough to produce a basin at the nest location. However, when the views are taken inside the cylinder constellation, homing vectors yield the agent away from the nest, like the Bn-models. Hence, the number of snapshots kept in memory, the locations where they are taken, as well as the cue they encode (brightness or CwN) are crucial parameters for the homing success of the model. Thus, we could select three models to be tested in conflict situations: the ALV, the B8-model and the CwN4-model.

Also the altitude at which the different homing models are tested influences their homing performance. At 6cm altitude both tested multi-snapshot models could predict the homing behaviour correctly, if a sufficient number of snapshots is taken. However, at an altitude of 15cm (see Methods for details) only the model using four or eight CwN snapshots predicts homing at the nest hole location in the non-conflict situation (S2 Fig). In contrast, both tested versions of Bn-models create several basins at a more central places in the arena. This result implies that the agent is likely to home at a location different from its nest and suggests the CwN-models to be more robust than the Bn-models.

Overall, the nature, number, spatial distribution and altitude of the views kept in memory does play a role in the homing success of multi-snapshot models during the non-conflict situation in our flight arena.

## Bumblebees' homing behaviour versus model performance during visual conflict

To predict the homing behaviour of bumblebees under cue-conflict conditions, we studied the performance of different homing models during visual conflicts. We did this only for the three models that were successful under the non-conflict condition at 6cm altitude, i.e. with a lowest isohypse surrounding the nest.

All models behave differently to each other during conflict situations. The ALV systematically leads the agent to a compromise location somewhere between the fictive nest holes of the two cue constellations and forms only one basin. This basin never indicates one of the fictive nests, as shown by the vector field of homing vectors and its potential (Fig 6B). For most conflict angles the agent is driven in between the cues. However, for a conflict of 90 degrees the agent is driven toward a location rather close to the cylinders.

For the multi-snapshot model, based on a brightness set of views taken outside the cylinder constellation (B8-model), (Fig 6C) basins are formed at locations where bumblebees search for their nest. The broad basin located at the stripes' fictive nest hole fits the behaviour of the bumblebees. However, the predicted basin at the cylinders is not always seen in the behaviour (e.g., -45/180). This result is reflected by an average F1-score close to 0.5 (0.45+-0.17) (Fig 7)). This intermediate F1-score is mostly due to a large number of false-positive predictions (S3 Fig) at the cylinders' fictive nest (e.g., -45/180), or due to a prediction of too large search areas (e.g., -45/45).

Fig 6D shows the results for the CwN4-model. This model has basins at the two fictive nest holes. Basins at the stripes' fictive nest hole are where the bumblebees appear to spend most time Fig 6D. In addition, the model does not create too many false-positive search predictions at locations different from the fictive nest locations (S3 Fig). The average F1-score for all conditions, of 0.52 +/- 0.15 thus reflects a slightly better performance than the other models tested (Fig 7).

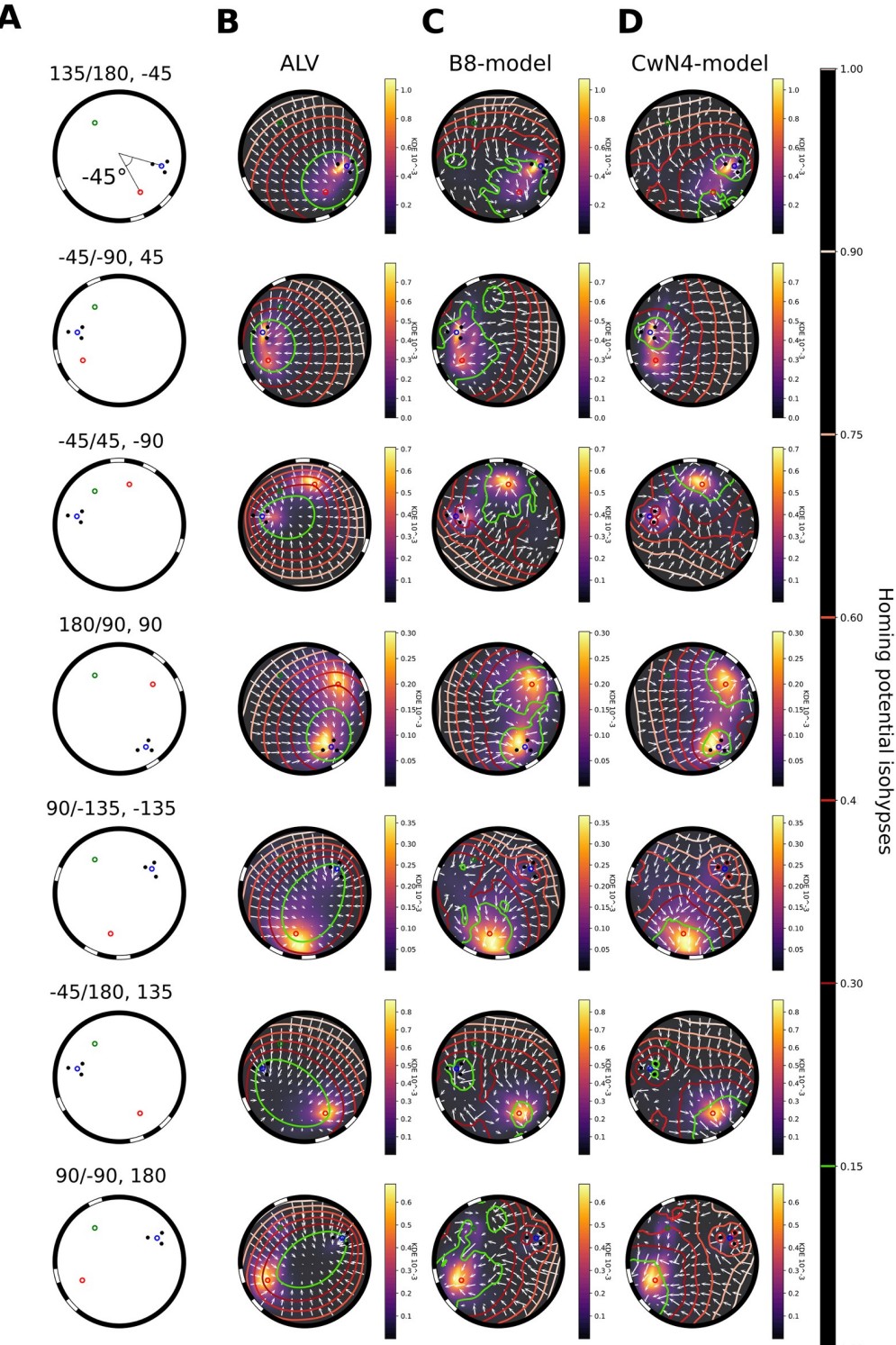

**Fig 6. Qualitative models and behaviour comparison for different conflict conditions.** A: Schematic of the set-up for the different conflict conditions. (BCD) The behaviour in the arena along the x and y axis of the arena is represented as a colour coded KDE from dark low distribution to yellow, high distribution. The vector fields of homing vectors for each model is represented on a 8-times downsampled grid by white arrows. Models results are expressed by an homing potential obtain from the Helmholtz-Hodge decomposition represented as contour lines or isohypses. The isohypse at 0.15 (basin) is coloured in green for clarity then the upper ones are coloured from dark red to white ("summit"). B: The ALV model for the 7 conditions, C: the B8-model, D: The CwN4-model.

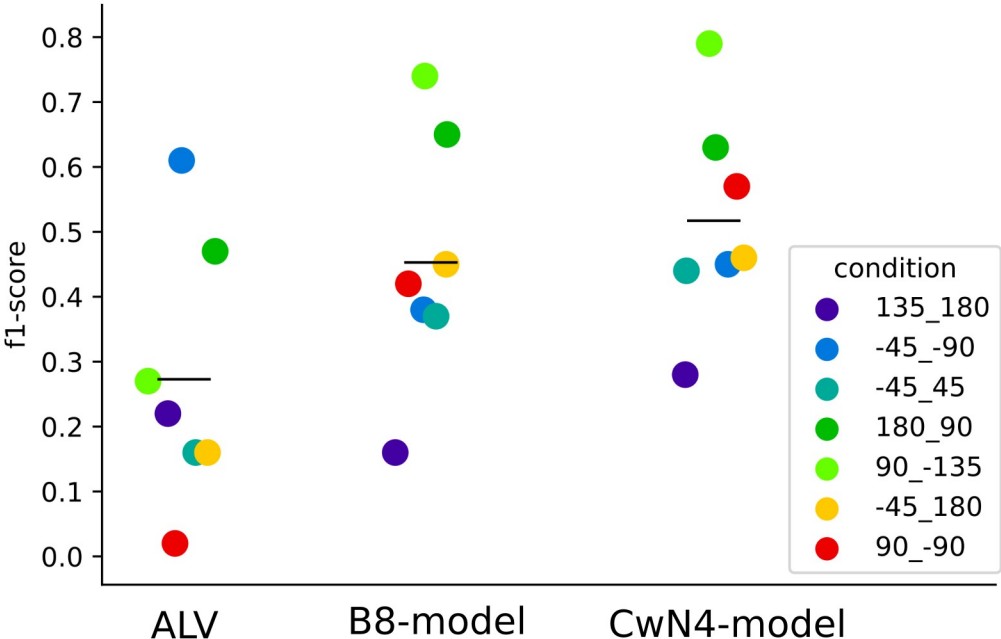

**Fig 7. Quantitative comparison of models and behaviour during conflict situations.** Distribution of F1-scores indicating the accuracy of the models prediction for each conditions describe with a colour code. The horizontal black bars represent the mean of the F1-score per model.

We finally wanted to test the robustness of the calculated F1-score by investigating if the selected isohypse, defining the search area of the model, has an impact on its performance in predicting the behavioural results. For the different conditions and models, it appears that a higher isohypse value and, thus, a larger predicted search area, is in general associated with a poor F1-score, except for the ALV model under some conditions (90/-135, -45/180). In contrast, a lower isohypse corresponding to a smaller predicted search area improves the score of the B8-model. The impact of the variation of the isohypse value on the CwN4-model is minor, making conclusions about its F1-score robust to variations of this parameter.

The quantitative analysis of the performance of the different models versus the bumblebees' search locations by varying the "lowest" basin height (S4 Fig) reveals a tendency for the CwN4-model to give a better prediction of the behavioural results for some of the cue-conflict conditions than the B8-model. Nevertheless, in most cases, this difference is not significant (three significant conditions out of seven: -45/45, 180/90, 90/-15, -45/180: $p > 0.5$; 135/180, -45/-90: $p < 0.001$, 90/-90: $p < 0.05$, Kruskal-Wallis posthoc Dunn's test, with Bonferroni correction for multiple comparisons). In conclusion, the two multi-snapshots models perform similarly to each other and, most important, likewise to the search behaviour of bumblebees as analysed under visual cue conflicts. However, the accuracy of the CwN4-model is slightly better than the B8-model in specific conditions.

## Discussion

### Bumblebee behaviour in an ambiguous visual scenario

Returning home after a foraging trip is, in addition to an innate aspiration to survive, a duty to ensure the growth of the colony [57]. Bumblebee foragers are excellent navigators and show astonishing homing abilities. In our study, bumblebees learned to return home in an artificial

visual environment providing them with two constellations of visual cues: small cylinders in the nest vicinity, and a stripe pattern on the wall of the arena. Our experiments clearly show the dedication of bumblebees to return home. Within the habitual visual arrangement of cues, our bumblebees flew straight back to the nest. However, after the environment has been visually altered, the bees obstinately searched for their home at the fictive nest entrances indicated by the different visual constellations. They did not search at a compromise location between the two fictive nest holes. This directed behaviour confirms the dominant role of vision in the context of local homing and could be used to distinguish between the explanatory value of different previously proposed models of homing mechanisms.

The bumblebees searched at the two locations indicated by the two landmarks constellations with a different probability depending on the degree of conflict between them. When the conflict was low, i.e. when the fictive nests indicated by the two cue constellations were close to each other, both fictive nest locations were visited. On the other hand, when the conflict was larger, for example when the constellations were opposite to each other, the bees almost only searched at the location indicated by the stripes. Bees hardly ever tried to find the nest hole at a compromise location between the two fictitious nest holes. In this respect, a non-holistic model such as the ALV could not predict this behaviour because it systematically led the agent to a compromise between the fictive nest holes.

Yet this model was not the only one failing to reproduce the homing behaviour characterised in the present study. The two holistic models we tested based on only a single snapshot taken at the nest location were unable to guide the agent home even in the non-conflict condition. What could be the reason for this performance?

## Homing based on only one snapshot

Numerous variants of homing models have been developed in recent years. The simplest variants of these models are based on memorizing a single snapshot taken at the nest site. Looking for a computationally parsimonious model to reproduce the search behaviour of bumblebees as characterized in the present study, we started with the classical homing model based on one brightness-snapshot (B1-model). This model gives a field of homing vectors and a corresponding homing potential, which is relatively shallow. This implies that the simulated bee flies on a random trajectory rarely leading to the nest hole (Fig 5). This model is still able to guide the agent to its home, but only when it is already close to the entrance, specifically when the agent is already inside the constellation of the three cylinders. Previously, this model has been used to explain homing in natural outdoor scenarios [30]. As the visual environment in our arena is much simpler than most natural environments, the failure of this model under these conditions may be due to the lack of visual disparity. Our results are consistent with previous indoor studies, where models based on a single snapshot image and a difference function did not fully predict the animals' return behaviour [34, 58]. In addition, there could be a geometrical reason why one snapshot is insufficient to reproduce the success of bumblebee homing in our installation: the third stripe is mostly occluded in the snapshot stored at the nest location (Fig 1D) with the consequence that an element of the cue constellation is missing in the memory.

## Homing based on several snapshots

Object occlusion is not only an issue in our artificial experimental setting (see Fig 2) but may well occur in more complex environments such as natural ones. One way to cope with the problem of occlusion is to use several snapshots at different locations around the goal location. Accordingly, several snapshots should reduce the likelihood of landmark occlusions as it was already suggested in previous studies [36, 43, 59, 60].

A brightness-based multi-snapshot model suggested by Graham et al 2010 was reported to guide an agent in cluttered environments [36, 43]. We found that this model was successful in guiding the simulated bee in our non-ambiguous situation. The success of this model depends on two intuitive parameters: the number of views memorised and their locations. Eight snapshots located around the nest hole outside the cylinder constellation allowed homing without any local-minima in other locations of the arena. However, this model failed if, for views taken at a higher elevation, the number of snapshots was only four, or if the snapshots were taken within the cylinder constellation.

The failure of the Bn-model when views are taken inside the cylinder constellation might be explained by the disparity between images being too small to guide the agent home (S5 Fig, Fig 5) and, hence, the four views sharing almost the same information. The impact of view disparity on the performance of homing models has been already discussed by Dewar et al. 2014. From a systematic analysis of the properties of sets of views, they observed that the homing performance was slightly affected by the level of disparity between views.

Nevertheless, we found that the B8-model, can predict the search location of the bumblebees in various visual-conflict situations. Since its predictions depend on where the views have been acquired, the question arises where bumblebees acquire the snapshots. On their first exits from the nest, naïve individuals perform a characteristic sequence of flights, which were shown in honeybees to be the basis of visual learning [61]. These learning flights start in the close vicinity of the nest hole and then the insect incrementally increases the distance from it in a loop-like manner [13, 48, 50, 51]. Hence, it is likely that the memorised views are taken by the bee within a radius of some twenty centimetres from the nest hole.

## Homing when distance information is included

Most variants of homing models rely on the brightness of the scenery. However, insects encode in their neuronal pathways not only the brightness and contrast of their environment, but also the apparent motion induced on its eyes by their ego-movement. These motion-dependent representations of the environment have been shown by model simulation to be correlated to the nearness of environmental objects weighted by their contrast [53, 54]. Distance information has been many times proposed as a plausible add-on to increase homing performance [41, 42, 51, 62–66], but never investigated so far with a multi-snapshot paradigm. The proposed CwN snapshot is a simple solution to the problem. The CwN may directly be obtained from what is perceived by the peripherical visual pathway via elementary motion detection mechanisms. In addition, it does not require complex computations much beyond local motion detection to be performed on the visual input [53, 54]. Finally, these CwN-views could be used in a similar way to the previous brightness-based views.

We found that the CwN4-model predicted the search locations of the bee in the tested cue-conflict situations in a more parsimonious way than the corresponding brightness-based model, i.e. with only four instead of eight snapshots. In addition, this CwN4-model could be shown to be more robust against changes in altitude than the Bn-models. This makes this model ecologically relevant as robustness against altitude changes is likely to be crucial for flying bees which perform learning flights at different heights and home by gradually lowering their altitude [51]. In conflict situations the CwN4-model also mimicked the preference of the bumblebees for the stripes' fictive nest when the cue conflicts were large (Fig 6D).

The good performance of the CwN4-model is likely due to its encoding of distance information on the one hand, and in addition, due to the highlighting of the edges of the different nearby objects in the environment (Figs 1E and 3C). This kind of representation of contrast edges is likely to underlie diverse behaviours of insects, like the use of edges for pattern

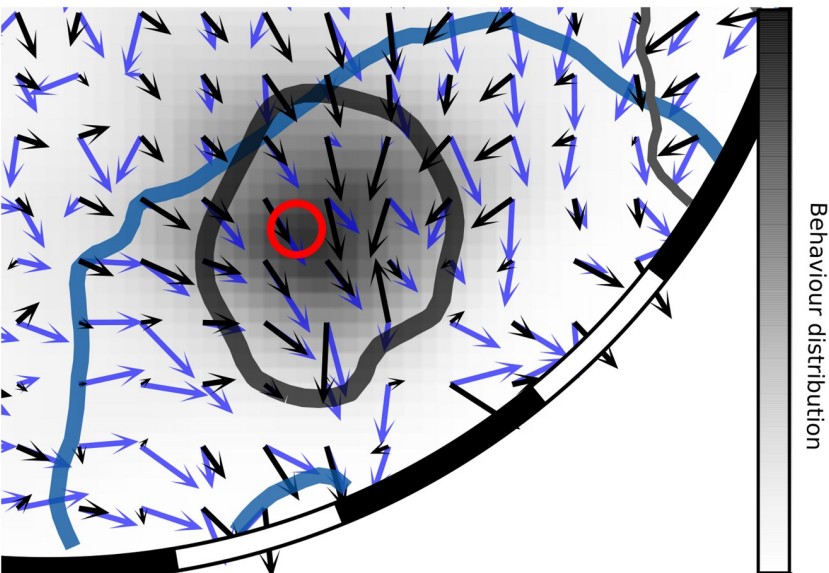

**Fig 8. Details of the field of homing vectors for the CwN4-model and for the B8-model.** The CwN4-model's vector field is represented in blue, with the lowest isohypse of its derived potential in dark blue. The B8-model's vector field is in black with the lowest isohypse of its potential in black. The distribution of the behaviour is represented by a black gradient from white to black. The red circle indicates the location of the stripes' fictive nest.

recognition and learning in bees [28, 67]. A close look at the homing vector directions determined with the CwN4-model shows that the agent is drawn to the edges of the stripes (Fig 8): this behaviour was observed also in real bumblebees (flight example, video, S1 Video). This strong edge attraction implies that this model cannot be the only mechanism involved in local homing in the close vicinity of the nest. The use of brightness at this point might give a more accurate home direction very close to the nest (Fig 8).

## Perspective on other models enabling homing without view rotations

On its way home, the agent must compare its memory with its current view of the environment. From our five tested models, only the ALV model proposed a comparison method which does not rely on the rotation of one of the sceneries against an other. A rotation of panoramic visual views can be accomplished in two ways: (1) as a physical rotation of the animal: the insect stops and scans its environment, such as ants do [45], or (2) as a mental rotation: the agent or insect rotates the panoramic visual representation of the environment in its brain. The first method (1) is not observed in bumblebees which mostly fly without interruptions and without rotating on the spot. For the second method (2), there is no evidence that it is feasible for insects, especially if several snapshots are required to be rotated. Yet other models, relying on information acquired at the nest entrance, have been proposed to enable homing without requiring a mental rotation. For example, the model proposed by Müller et al. (2018) which does not store the full panoramic image at the goal location but only a panoramic skyline [27], from which, as the ALV, a simple vector is memorized to enable homing. Möller et al 2001, have suggested a model descending a gradient encoding for a measure of landmarks' contours to guide homing [37]. Stürzl et al. 2016 have proposed a model using Fourier transforms of the views and comparing them based on their phases [41]. Stone et al. 2018 suggested a model based on skyline- and rotation-invariant information by using spherical harmonics

[26]. And finally, a recent model using an oscillator and the combination of attractive (oriented toward the nest) and repulsive views (oriented away from the nest) works without stop phases [38].

From all listed models, only the model proposed by Le Möel et al. 2020 [38], uses several snapshots to guide the agent home. Hence, the other mentioned models [26, 27, 37, 41] are likely to suffer from object occlusion as the B1 or CwN1 model do. Therefore, their performance is expected to be similar to that of the B1 or CwN1 model in the non-conflict situation.

At last, since the model designed by Le Möel 2020 relies on numerous snapshots, it is likely to predict the bumblebees' behaviour in conflict-situations such as the Bn or CwNn models do. Therefore, it would be of interest to study the performance of this model during return flights and compare it with the behaviour of the bees.

## Conclusion

When bumblebees return to their nest, they are primarily guided by visual cues. When cues are placed into conflict indicating two possible nest locations, bees search at these and not at some intermediate locations. Holistic models using several views could reproduce the observed homing behaviour of the bees. The success of the different homing models in reproducing the search location of the bees in an ambiguous scenario could be attributed to two aspects: (1) storing multiple snapshots of the home location outside a constellation of local landmarks and (2) plausibly encoding information about the 3D layout of the environment. The multi-snapshot model based on views encoding spatial information via the contrast-weighted nearness provided by the motion detection system covers both aspects. This model brings a simple explanation to an apparently complex behaviour. Eventually, this model could easily be implemented on a technical agent using biologically inspired motion perception system for navigation, without requiring additional equipment.

## Supporting information

**S1 Fig. Trajectories examples and illustration of the behavioural variability.** For each condition, we represented 3 trajectories examples. The examples were selected based on the amount of time the bees spend searching. Each trajectory is colour coded. The longest flight trajectory is in blue, the shortest in red and the average length trajectory for this condition ín black.
(PDF)

**S2 Fig. Homing potential of all different models during non-conflict situation at 15cm elevation.** The ALV model is not affected by the change in altitude since it uses the exact position of the cues in the 2D plane.
(PDF)

**S3 Fig. 2D representation of the confusion matrix values for each conflict condition.** Each subplot represents the confusion matrix output for each model during tested conflict conditions, from left to right: ALV, B8-model, and the CwN4-model. Each title informs the tested condition followed by its F1-score. The colours describe the correct predictions: the true negatives in orange and the true positives in yellow, while the failure of the model predictions are: false positives in purple and the false negatives in dark blue.
(PDF)

**S4 Fig. F1-score when varying the model prediction isohypse.** Each plots represents for each condition the F1-score depending on the selected isohypse from 0.1, pink, to 0.29, light brown. A Dunn's Post-hoc test following the Kruskall-Wallis test was performed on each condition:

the adjusted significance values are represented when significant. The significance levels are coded as follow: p<0.5 *, p<0.1 **, p<0.01 **, p<0.001 ***, p<0.0001 ****.
(PDF)

**S5 Fig. The set of views *S* for the B4-model taken at 5cm from nest hole.** The 4 views represented as an equirectangular projection is overlaid with the corresponding rotIDF function with the current view as in Fig 2B.
(PDF)

**S1 Video. Bumblebee attracted to the stripe edges.**
(AVI)

## Acknowledgments

We thank Tim Siesenop and Ronja Bigge to have helped with the experiments as well as Meike Puts and Marina Inhofer for the long reviewing of the bumblebees' trajectories. We also thank Sridhar Ravi for his support at the beginning of the project. In addition, we acknowledge the work of Luise Odenthal, who conceived many test-functions for the different models. Finally, we thank Thierry Hoinville to have provided and introduced us to the Helmholtz-Hodge decomposition and for his helpful scientific remarks on the early draft.

## Author Contributions

**Conceptualization:** Charlotte Doussot.

**Data curation:** Charlotte Doussot, Olivier J. N. Bertrand.

**Formal analysis:** Charlotte Doussot.

**Funding acquisition:** Martin Egelhaaf.

**Investigation:** Charlotte Doussot.

**Methodology:** Charlotte Doussot, Olivier J. N. Bertrand.

**Software:** Charlotte Doussot, Olivier J. N. Bertrand.

**Supervision:** Olivier J. N. Bertrand, Martin Egelhaaf.

**Writing – original draft:** Charlotte Doussot.

**Writing – review & editing:** Charlotte Doussot, Olivier J. N. Bertrand, Martin Egelhaaf.

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
