## [Decision Letter · Decision Letter 0]

22 Mar 2020

Dear Mrs Doussot,

Thank you very much for submitting your manuscript "Comparative study of visually based homing models during visual conflict" for consideration at PLOS Computational Biology.

You will see below that two highly skilled reviewers assessed your manuscript. Both see merit in the study and potential impact in the field of insect navigation, and so do I. The reviewers also raised constructive major concerns that I think can be addressed with major revisions. I hope you will find these comments useful.

We cannot make any decision about publication until we have seen the revised manuscript and your response to the reviewers' comments. Your revised manuscript is also likely to be sent to reviewers for further evaluation.

Sincerely,

Mathieu Lihoreau, Ph.D

Guest Editor

PLOS Computational Biology

Wolfgang Einhäuser

Deputy Editor

PLOS Computational Biology

Dear Dr. Doussot,

Thank your for submitting to PLoS Comp Biol. You will see below that two highly skilled reviewers assessed your manuscript. Both see merit in the study and potential impact in the field of insect navigation, and so do I. The reviewers also raised constructive major concerns that I think can be addressed with major revisions. I hope you will find these comments useful.

All the best,

Mathieu Lihoreau

Reviewer's Responses to Questions

**Comments to the Authors:**

Reviewer #1: The authors trained bumblebees to navigate in a large cylindrical arena between the entrance of their nest (on the floor of the arena), and a feeder accessible through a small hole in the top part of the arena wall. The arena was designed to provide visual cues (3 stripes on the arena hole) and three cylindrical object standing on the floor around the nest entrance. The bumblebees homing search were analysed during tests were the two sets of cues were rotated relatively to each other, providing visual conflicts. In parallel, the authors implemented different visual-based homing models using reconstructed views of the arenas visual cues. Modelled agents were given some memories of the training situation, and tested in the different visual conflict situations. The authors calculated a fit (f1 score) for each model by comparing the likelihood where the agents were to end their journey (using Helmholtz-Hodge decomposition) to the spatial distribution of the bumblebees’ search, for each conflict situation. The model that obtained the highest overall fit (average f1 score across test situations) was equipped with a ‘Contrast-weighted Nearness’ visual front end views, that is, its visual input encodes both luminance and information about depth. The author concludes that encoding depth information (presumably through optic flow) form a ‘biologically plausible hypothesis’ (or ‘biologically inspired strategies’) to solve this kind of navigational tasks.

I first have to say that the bumblebee experiments, the models’ implementation, and the analysis are all quite ingenious, properly achieved and well described despite their necessary complexity. The figures are very neat and the results are well presented. This reflects a rare combination of high capability for both behavioural experiments and modelling.

I have however concerns about the more fundamental scientific approach used here.

It is not clear to me actually what is the scientific question here? Why did the authors choose to design these particular visual conflict situations? And why did they choose this 4 particular models with these specific set of parameters? The conclusions are based on the fit between the models and the bumblebees search, but it seems quite clear that this fit (and thus the conclusions) will be sensible to: 1- changing the parameters of the models (for instance, what if the number and position of the memorised views are different, what if the eyes are not spherical anymore? Etc..), 2- changing the type of visual environment (for instance, an open vs cluttered natural scene would change the usefulness of depth information) or the type of visual conflict situation in this arena (what if the cylinders are removed?), 3- the way the bumblebee and model’s search is measured for comparison (what if, instead of ‘likelihood of ending up’, we look at the body orientation and straightness of paths ? Perhaps in that case the AVL (currently the poorest fit) would obtained the best fit, because to derive a direction the multiple snapshot models require to stop and scan at each time step (bumblebee don’t do that?) and the gradient descent or ascent require first to sample the neighbour locations (bumblebee don’t do that?). 4-the very choice of models (is there a reason for not testing a visual parameter based model (Möller 2001) for instance?).

Möller, R. "Do insects use templates or parameters for landmark navigation?." Journal of Theoretical Biology 210.1 (2001): 33-45.

If that is right, this makes your conclusion quite weak.

Currently, it feels like the modelled agents are used here as hypotheses themselves (which, as explained above, make the conclusions dependant on the specific parameter-agent-environment-measurements chosen), rather than as a tool to proof-test the viability of a specific hypothesis, or as a tool to explore the information available in environments. I exemplify the two latter approaches below.

The main conclusion here seems to be about the 'plausibility' and 'usefulness' of depth information, but these notions seems quite unrelated to the overall approach. If the question is about ‘the plausible use of depth through optic flow in bumblebee’, a definite answer can be obtained with a dedicated experiment, such as testing bees with landmarks that are camouflage against the background (as achieved previously by this lab in the past with bees (Dittmar et al. 2010)). Here a model can be used as proof-test that the camouflage landmark can be revealed using optic-flow as long as you do translational movement. If the question is about 'the usefulness of depth information for navigation in flying agents’, it can be tackled with a systematic analysis of the information provided by depth in natural scenes, and agents can be used to estimate the usefulness of this information for navigation (as achieved previously Stürzl, W. and Zeil, J. 2007).

Dittmar, L. et al. "Goal seeking in honeybees: matching of optic flow snapshots?." Journal of Experimental Biology 213.17 (2010): 2913-2923.

Stürzl, W. and Zeil, J. "Depth, contrast and view-based homing in outdoor scenes." Biological cybernetics 96.5 (2007): 519-531.

I would thus recommend to clearly establish a question together with hypotheses, and explain how the models and behavioural experiment will ‘a priori’ enable to disentangle the hypotheses and thus answer the question. Alternatively, it could be stated that the current study is not hypothesis driven but purely exploratory (but this may not be particularly helpful as such explorations are generally useful when achieved in settings were the observer wants to diminish the influence of its own assumptions) in that case it would require at least a detailed analysis of the bumblebees, and a detail analysis of models’ behaviour across their parameter space, and model should be chosen so as to embed general concept about either visual processing (what the nature of the visual information used (e.g., depth vs no depth)) or sensori-motor processing (how this information (whatever its nature) is transformed into action (e.g., gradient descent vs. multiple snapshot)), but not both at the same time.

I hope my comments will be perceived as constructive, and apologise if there has been misunderstanding on my side.

with best wishes,

Reviewer #2: Comparative study of visually based homing models during visual conflict

General Feedback:

I think this is an interesting study that combines new behavioural results with modelling analysis to advance our understanding of how insects pinpoint inconspicuous locations like a small nest entrance. As such it is well suited to the readership of PloS Comp Bio.

I have no major concerns about validity of the behavioural or modelling works but think that the paper needs some series editing before I can recommend for publication in PloS Comp Bio. Below I list my main concerns and where I feel it appropriate offer some opinions on how they might be addressed.

Introduction:

Here is where I have some of my most series issues with the current manuscript.

Firstly sufficient care has not been taken to cite original papers. For example, for route following in ants you should be citing Kohler and Wehner, 2005 and Mangan and Webb, 2012 rather than paper you currently cite which instead speaks about temporal information for place recognition (Graham and Mangan, 2015). There are other examples.

Secondly, if you are to perform a comparative study of models there is a clear need to (a) provide comprehensive coverage of what models, or at least model categories, have been proposed and their strengths and weaknesses. This is currently lacking with many models missing e.g. from Jochen Zeil, Wolfgang Sturzl and recent models using frequency cues e.g. Stone et al, 2018 "Rotation invariant visual processing for spatial memory in insects". In this regard, there have been a number of comparative studies on visual homing implemented previously which are missing from the current text e.g. Vardy A (2005) Biologically plausible methods for robot visual homing. PhD thesis, Ottawa-Carleton Institute for Computer Science"; Basten Mangan and Webb, 2009 Modelling place memory in crickets; and even Wystrach et al, 2013 "Snapshots in ants? New interpretations of paradigmatic experiments". Indeed, Jochen Zeil's 2012 paper: "Insect Homing: an insect perspective" is highly relevant here. Note, on the above comment, the research of Jochen Zeil focusses on the final approach of digger wasps to their inconspicuous nest entrance and appears particularly relevant here. I recommend a thorough redraft of the literature review to set the context for the reader.

This leads directly to my next concern. What motivated your choice of model to implement and test? This is not clear to me from the current paper. If this is to be a truly comparative study, then you should be testing across a range of models, or model categories, that give a good representation of the current hypotheses from the field. Otherwise, it can appear as a straw man argument to support your preferred model. Note: I am not suggesting that you reimplement models to add to your data but instead describe your criteria for model selection and then discuss how your results gathered on a sample of models impact current theories.

If I may offer some advice it seems to me that you are not focussed enough on testing a specific hypothesis. I would suggest that two recent hypotheses arise from the works of Dewar (as you refer to later in your paper) who proposes that by aligning with multi nest facing views one can return home, and Sturzl et al, 2016; Le Moel, 2020 that propose instead having mutliple repulsive view leads to more robust homing. In essence, your study is testing the validity of Dewar's proposal using a visual distortion (not a cue conflict scenario) paradigm and your results support his work. Thus, I would restructure the Intro to focus on this key issue and work from there.

Behavioural Study:

The data presented gives n=4 example flights but then search distributions for n=5-10 flights. What I am not sure about, and cannot find explained in the manuscript is how this data is made e.g. is this one track from 4 different bees or multiple tracks from the same bees. This is very important and should be clearly stated.

I think the search patterns shown are an excellent way to show the data but I think Fig 4 could do with more labelling and also some additional clarity. For example, the reader has to constantly look back to A to recall where the landmarks were in training to compare with the new results. Also, they need to decode what each title means. You could add an outer ring to each figure showing the location of landmarks in training and a labelled arrow showing that rotation. Similarly, you could show the location of the landmarks as white dots in each figure and a similarly labelled arrow to show that rotation. That would make it easier for readers to understand.

Modelling study:

For the final approach, there are two models describing how views could be used in a 'repulsive' manner to drive approach. "W Stürzl - ‎2016, How wasps acquire and use views for homing"; "" Opposing views method. Le Möel, 2020 "Opponent processes in visual memories: A model of attraction and repulsion in navigating insects’ mushroom bodies". How would this compare?

Do you not have data on the learning flights which they could add to the models? Are these structured that make one model in particular better? I think adding this element to the modelling work would be very interesting and increase the impact a lot.

Why include the results for the ALV at all? This model has already been shown unable to account for animal data numerous times and just confuses the matter. The benefits of multi views is for me the main contribution. I am not sure that the reader gains anything from the inclusion of the ALV data.

Specific Comments

Terminology - there is repeated use of the phrase "to find back home" which is strange in English. I would replace with either "to return home", or to "to find home"

Table 1. should memory for CWN multi-snapshot not be "several CwN snapshots"?

**Have all data underlying the figures and results presented in the manuscript been provided?**

Reviewer #1: Yes

Reviewer #2: Yes

PLOS authors have the option to publish the peer review history of their article (what does this mean?). If published, this will include your full peer review and any attached files.

Reviewer #1: Yes: Antoine Wystrach

Reviewer #2: No
---

## [Editor Report · Decision Letter 1]

19 Aug 2020

Dear Mrs Doussot,

We are pleased to inform you that your manuscript 'Visually guided homing of bumblebees in ambiguous situations: a behavioural and modelling study' has been provisionally accepted for publication in PLOS Computational Biology.

Best regards,

Mathieu Lihoreau, Ph.D

Guest Editor

PLOS Computational Biology

Wolfgang Einhäuser

Deputy Editor

PLOS Computational Biology

Your manuscript has been reviewed by two highly skilled experts. I shared most of their concerns that you convincingly addressed in this revised version. This is a nice piece of work, well done!

---

## [Editor Report · Acceptance letter]

28 Sep 2020

PCOMPBIOL-D-19-02024R1 

Visually guided homing of bumblebees in ambiguous situations: a behavioural and modelling study

Dear Dr Doussot,

I am pleased to inform you that your manuscript has been formally accepted for publication in PLOS Computational Biology. Your manuscript is now with our production department and you will be notified of the publication date in due course.

With kind regards,

Matt Lyles
